# The effect of a task-specific training on upper limb performance and kinematics while performing a reaching task in a fatigued state

Frédérique Dupuis[1,2], Félix Prud'Homme[1,2], Arielle Tougas[1,2], Alexandre Campeau-Lecours[1,2,3], Catherine Mercier[1,2], Jean-Sébastien Roy[1,2]*

1 Faculty of Medicine, Université Laval, Quebec City, Quebec, Canada, 2 Centre for Interdisciplinary Research in Rehabilitation and Social Integration (Cirris), Quebec City, Quebec, Canada, 3 Faculty of Science and Engineering, Université Laval, Quebec City, Quebec, Canada

* Jean-Sebastien.Roy@fmed.ulaval.ca

## Abstract

### Background

Fatigue impacts motor performance and upper limb kinematics. It is of interest to study whether it is possible to minimize the potentially detrimental effects of fatigue with prevention programs.

### Objective

To determine the effect of task-specific training on upper limb kinematics and motor performance when reaching in a fatigued state.

### Methods

Thirty healthy participants were recruited (Training group n = 15; Control group n = 15). Both groups took part in two evaluation sessions (Day 1 and Day 5) during which they performed a reaching task (as quickly and accurately as possible) in two conditions (rested and fatigued). During the reaching task, joint kinematics and motor performance (accuracy and speed) were evaluated. The Training group participated in three task-specific training sessions between Day 1 and Day 5; they trained once a day, for three days. The Control group did not perform any training. A three-way non-parametric ANOVA for repeated measures (Nonparametric Analysis of Longitudinal Data; NparLD) was used to assess the impact of the training (Condition [within subject]: rested, fatigued; Day [within subject]: Day 1 vs. Day 5 and Group [between subjects]: Training vs. Control).

### Results

After the training period, the Training group significantly improved their reaching speed compared to the Control group (Day x Group p < .01; Time effect: Training group = p < .01, Control group p = .20). No between-group difference was observed with respect to accuracy. The Training group showed a reduction in contralateral trunk rotation and lateral trunk

**Data Availability Statement:** Data cannot be shared publicly because of ethical restrictions. Data are available from the Lyne Martel, Ethics

Committee (contact via lyne.martel2. ciussscn@ssss.gouv.qc.ca) for researchers who meet the criteria for access to confidential data.

**Funding:** This study was funded by the Natural Sciences and Engineering Research Council of Canada (RGPIN-2023-04929). Frederique Dupuis is supported by a scholarship from the Canadian Institute of Health Research (CIHR). Jean-Sebastien Roy is supported by salary awards from Fonds de recherche Québec – Santé (FRQS) and Catherine Mercier holds the Canada Research Chair in Sensorimotor Rehabilitation and Pain and the University Laval Research Chair in Cerebral Palsy. The funders had no role in study design, data collection and analysis, decision to publish, or preparation of the manuscript.

**Competing interests:** The authors have declared that no competing interests exist.

flexion in Day 2 under the fatigue condition (Group x Day $p < .04$; Time effect: Training group = $p < .01$, Control group = $p < .59$).

## Conclusion

After the 3-day training, participants demonstrated improved speed and reduced reliance on trunk compensations to complete the task under fatigue conditions. Task-specific training could help minimizing some effects of fatigue.

## Introduction

In recent years, there has been a growing interest in understanding the phenomenon of fatigue. Fatigue is a common experience in everyday activities and work, with potential consequences such as decreased performance, and reduced productivity [1–3]. It is recognized as a symptom that affects both physical and cognitive functions, leading to alterations in muscle function and the central nervous system's ability to plan voluntary movement, and ultimately resulting in changes in movement control [4, 5]. These changes in movement control can increase the risk of injuries by impacting the mechanical loads on musculoskeletal structures, such as tendons, muscles, and cartilage [6]. The shoulder joint, being the most mobile joint of the body, is particularly vulnerable to fatigues, as its stability highly depends on neuromuscular control [4, 7].

Fatigue has been extensively studied as a symptom [4]; however, it remains crucial to identify strategies for preventing its detrimental consequences, especially for individuals exposed to repetitive tasks. If these detrimental consequences are associated with changes in motor control, one potential approach could involve implementing prevention interventions. In these adaptations to fatigue, evidence suggests involvement of various levels of the motor system, spanning from the spinal cord to the motor cortex [6, 8]. Optimizing motor control during tasks that pose a high risk of injury, such as those involving elevated arm positions, could prove to be a promising preventive measure [9]. This could be achieved through motor training interventions that promote motor learning, entailing repeated practice of context-specific motor tasks [8]. Motor training has been primarily used in sport-injury prevention programs to enhance performance and reduce injury rates [8, 10, 11]. Additionally, motor training has shown effectiveness in improving muscle recruitment patterns, even in tasks outside the context of sport-injury prevention [12–14]. It is believed to improve movement planning, strengthen internal representation, and reinforce feedforward control [12–14].

Based on these underlying mechanisms of motor training, we hypothesize that task-specific motor training could mitigate the adverse effects of fatigue (i.e., upper limb kinematic alterations and decreased performance) [10, 11, 15, 16]. However, to the best of our knowledge, no study has yet investigated this potential preventive strategy. Therefore, the primary objective of our study is to explore the impact of task-specific training on upper limb kinematics and motor performance during a reaching task performed in an elevated position under a state of fatigue. The choice of an elevated position reaching task is based on the understanding that it poses a heightened risk of shoulder injury with repeated execution [17].

## Methods

Healthy participants were recruited between May 16[th] and June 30[th], 2022, and randomly assigned to either the Training group or the Control group. Participants were aged between 18

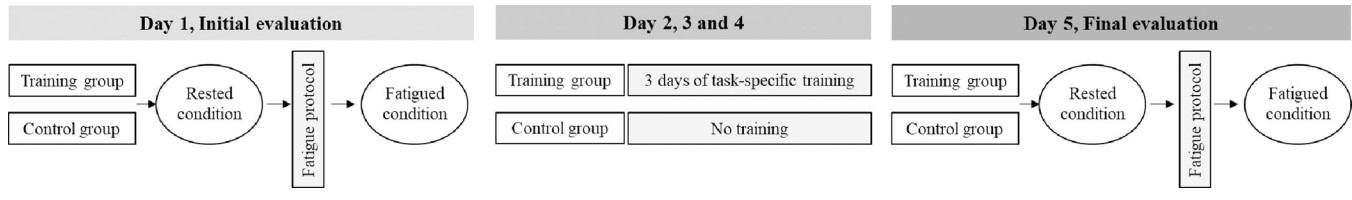

**Fig 1. Experimental design.**

and 30 years old, did not have any self-reported shoulder or neck pain/disability, and had no history of upper limb/spine fracture, surgery, or shoulder dislocation. The Sectorial Rehabilitation and Social Integration Research Ethics Committee of the CIUSSS-CN approved this study (2017–527) and written informed consent were obtained from every participant.

Both groups participated in two evaluation sessions, with a 5-day interval between them (i.e., Day 1 and Day 5). During these sessions, they performed the same reaching task under two conditions (rested and fatigued). Both evaluation sessions were conducted at the same time of the day, and all participants first performed the task in the rested condition, and then in the fatigued condition, after completing a shoulder fatigue protocol (Fig 1). During the reaching task, data on surface electromyographic (sEMG) activity, joint kinematics and motor performance were collected.

## Reaching task

The reaching task was performed in a virtual reality environment using Unreal Engine (Epic games international, Unreal Engine, Switzerland). Participants wore HTC VIVE goggles (HTC corporation, VIVEPORT, Taoyuan City, Taoyuan County, Taiwan) that exposed them to the virtual environment. The task consisted of reaching five different targets (i.e., 5cm red radius ball) located around the participants in the virtual environment. They performed the task in a sitting position, the trunk free to move, and only their feet were fixed on the ground. The participants were asked to move naturally. Using a goniometer, the five targets were positioned prior to the experiment as follow: Target 1 = 90° of humeral elevation in the frontal plane (abduction), with 90° of humeral external rotation and elbow flexed at 90°; Target 2 = 90° of humeral elevation in the frontal plane (abduction) with the elbow extended; Target 3 = 120° of humeral elevation in the plane of the scapula (scaption) with the elbow extended; Target 4 = 120° of humeral elevation in the sagittal plane (flexion) with the elbow extended; and Target 5 = 140° of humeral elevation in the sagittal plane (flexion) with the elbow extended (S1 Fig). To complete the task, they had to reach each target 5 times in a random order, for a total of 25 targets reached (average 60 seconds). Every reaching movement started from the same initial position (i.e., starting position). To standardize this starting position, an additional target (5cm radius ball) was positioned in front of the participant, at 90° of humeral elevation (in the sagittal plane, elbow extended). Participants were required to return to this target between each reaching movement to initiate the release of the subsequent target.

The instruction given to the participants were to reach the targets as quickly, but also as accurately as possible using the most direct way through the targets. The participants held a controller in their dominant hand; it appeared to the participants as a virtual hand. In this virtual hand, they could see a 2cm radius ball on the palm. Participants were asked to place the 2cm radius ball directly through the 5cm radius target to succeed the reaching movement.

The participants' perceived level of fatigue was assessed using the Borg Rating of Perceived Exertion Scale [18] both before and after each trial (10 points scale, 0 = no exertion and 10 = total exertion). One practice trial was performed before the beginning of the experiment.

## Fatigue protocol

On both evaluation sessions, all participants performed a fatigue protocol before their second trial. The fatigue protocol used in this study has previously been validated and detailed by Ebaugh et al [19]. It consisted of three different tasks completed with the dominant arm: 1) manipulating screws on a wooden board for 2 minutes with the shoulders at 45˚ of flexion; 2) 20 repetitions of arm elevations in the sagittal plane holding a dumbbell; and 3) 20 repetitions of arm elevations in the scapular plane holding the same a dumbbell. The dumbbell used were 0.9 kg for women and 1.8 kg for men. The participants rated their perceived level of exertion every 30 seconds using the Borg Rating of Perceived Exertion Scale [18]. The three tasks of the fatigue protocol were repeated until the participant reached a perceived level of exertion of at least 8/10. This protocol was chosen as it has been shown to lead to decreased motor performance and kinematics alterations during reaching, as indicated by a previous study [20]. Therefore, it enables the evaluation of the potential effects of the task-specific training on these adaptations.

## Task specific training

While the Control group only participated in the two evaluation sessions (Day 1 and Day 5), the Training group took part in three training sessions between Day 1 and Day 5 (training once a day during Day 2, 3, and 4). There is significant variability in the literature regarding motor training parameters targeting motor learning, but they typically involve task-specific practice with conscious attention to performing the appropriate movement [8]. Accordingly, during the training sessions, participants in the Training group performed the same task executed during evaluation sessions (task-specific training) [17, 18], five times using only the dominant arm. Altogether, the Training group completed the task 15 times between the two evaluation sessions, totaling 375 reaching movements. During a typical training session, they were instructed to focus on their accuracy for one trial (the most accurate performance without considering speed), on their speed for one trial (the fastest performance without considering accuracy) and then practice the combination of speed and accuracy for three trials. A minimum of two minutes rest periods were provided between each trial to prevent fatigue, which could be extended until participants rated their perceived level of exertion at 0 out of 10. The objective of the training was to let participants find and practice their own perceived effective strategy and reinforce their motor system ability to perform the task.

## Measurements and outcomes

Performance was assessed using Unreal Engine, enabling us to track participants' hand in a three-dimensional space with the controller. Performance data were extracted using custom software written in MATLAB. Performance data included reaction time, time to reach the targets and accuracy [2]. The reaction time was calculated from the moment the randomly released target appeared in the virtual environment to the moment the participant left the starting position to initiate the reaching movement. The time to reach the targets was calculated from the moment the participant left the starting position and reached the target (i.e., successfully placed the 2cm radius ball in the 5cm radius target). Accuracy was defined by 3 different variables. First, the initial angle of endpoint deviation (iANG) represented the initial trajectory of the hand. This angle was calculated using the shortest line between the starting position and the reaching target, and the line corresponding to the initial peak of acceleration [20]. It reflected movement planning, where a larger angle represents a larger error in movement planning. Second, the final error (fERR), measured as the arc distance between the ideal arrival point into the target reached (i.e., the most direct way) and the actual arrival point,

reflected reaching accuracy. Third, the area under the curve (area) was calculated to represent to total movement error while reaching. It was calculated as the difference between the ideal trajectory (i.e., most direct way) and the actual trajectory used by the participants in the 3-dimensional space. More precisely, it is the summation of the rectangular trapezoids perpendicular to the ideal trajectory line and the actual trajectory line [20]. The mean values of the 25 reaching movements were calculated for every trial for analysis.

The second variable of interest was upper limb and trunk kinematics. It included joint angles at the trunk, sternoclavicular joint, shoulder, and elbow. Joint angles were measured using six inertial measurements units (IMUs) (MVN, Xsens Technologies, Enschede, Netherlands), positioned in accordance with Xsens sensors configuration at the trunk, head, sternum, dominant scapula, and dominant arm (i.e., arm and forearm). The IMU data were acquired at a sampling rate of 100Hz with a custom Matlab IMU acquisition software. The latter requires a calibration sequence consisting of a static position (arms alongside the body) and dynamic movements (raising the arm, flexing the trunk). The IMUs data were imported into MATLAB R2018a (The Math Works Inc., Natick, MA, USA) and data fusion was performed with a custom algorithm to obtain joint angles [21]. To describe the reaching movement, initial angles and final angles during the movement were calculated. Initial angles represent the angles before the beginning of the reaching movement, while waiting at the starting position. It reflects initial posture. Final angles were calculated when the targets were reached. It reflects movement strategy to reach the targets. The movement of interest were flexion/extension, rotations and lateral flexions at the trunk, elevation/depression at the sternoclavicular joint, elevation, plane of elevation, and rotation at the shoulder, and flexion/extension at the elbow. Mean values of the twenty-five reaching movements were calculated and used for statistical analysis.

## Muscles fatigue assessment

To monitor the presence of fatigue when performing the reaching task in the fatigued state, wireless sEMG sensors (Delsys Trigno, USA) were placed on the anterior and middle deltoids and on the upper trapezius of the dominant arm. These muscles were chosen because they are the main agonists in shoulder elevation [20]. The skin was cleaned using alcohol prior to electrode placement, and the sensors were positioned according to The Surface EMG for Noninvasive Assessment of Muscles (SENIAM) [22]. Muscle activity was recorded using Delsys EMGworks[R] Acquisition software (sampling rate: 1925.93Hz). All sEMG signals were processed using custom software written in MATLAB R2013a (The MathWorks Inc., Natick, Massachusetts, United States). sEMG signals were digitally filtered off-line with a zero-lag 4[th] order Butterworth Filter (band-pass 20–450Hz) [20]. The power spectrum density was computed from the squared Fast-Fourier Transform. Fatigue was characterized as a downward shift in the sEMG power spectrum (i.e., median power frequency [MDF]), associated with an increase in sEMG signal amplitude [23].

## Statistical analyses

Characteristics of both groups were compared using independent t-tests and χ2. For all variables, including the MDF and the sEMG amplitude of each muscle, a Nonparametric Analysis of Longitudinal Data (NparLD) was conducted using a three-way non-parametric ANOVA for repeated measures (Condition [within subject]: rested, fatigued; Day [within subject]: Day 1 vs. Day 5 and Group [between subjects]: Training vs. Control) was used. NparLD analyses are particularly relevant for small samples and do not require normality of the data [24]. Non-parametric post-hoc analyses were conducted to detail the differences when a significant

**Table 1. Participants characteristics.**

| Characteristics | Training group (n = 15) | Control group (n = 15) |
|---|---|---|
| Age (years; mean +/- SD) | 26.3±3.5 | 24.1±3.2 |
| Sex (N female) | 8 | 7 |
| Dominance (N right-handed) | 15 | 15 |
| Weight (Kg; mean +/- SD) | 69.5±9.6 | 66.3±12.7 |
| Height (cm; mean +/- SD) | 171.9±10.7 | 171.9±9.9 |

interaction was present. Statistical analyses were conducted out using R 4.1.0 [24]. The significance level was set at 0.05.

# Results

Thirty participants were recruited and assigned to either the Training group (n = 15) or the Control group (n = 15). There was no statistically significant between-group difference (p>.05) for baseline characteristics (Table 1).

## Perceived level of exertion and fatigue protocol

Mean perceived level of exertion after completing the task during the rested condition at Day 1 was 2.5±1.0/10 for the Training group and 2.9±1.4/10 for the Control group. There was a significant decrease for both groups of the mean perceived level of exertion in the rested condition between Day 1 and Day 5 (Day effect p < .01). The mean ratings decreased to 1.9±1.1/10 and 2.1±1.9/10 for the Training and the Control group, respectively. The fatigue protocol was performed for a mean duration of 298 sec on Day 1 and of 320 sec on Day 5 and led to a mean perceived level of exertion of 8/10 on both days. There was no between-group difference for the fatigue protocol duration or level of exertion after the protocol. As expected, both groups significantly perceived the task more demanding after performing the fatigue protocol, during the fatigued condition, with a mean perceived level of exertion of 7.5±1.2/10 for the Training group and 7.9±1.1/10 for the Control group. However, the mean perceived level of exertion in fatigued condition did not change between days (p = .13), regardless of the group.

## Muscles fatigue assessment

After the fatigue protocol, significant EMG signs of fatigue were identified among the agonist muscles (Condition effect p < .01) as characterized by a significant increase in Anterior deltoid, Middle deltoid and the Upper trapezius sEMG amplitude and a significant decrease of their MDF on both days during the fatigued condition. There was no difference between the days of assessment, or the groups, on muscle fatigue (Time effect and Day x Group interaction p>.43).

## Motor performance

Performing the task in a fatigued state led to decreased motor performance. There was a significant increase of the iANG (Condition effect, p = .02), fERR (Condition effect, p < .01) and of the time to reach the targets (p < .01) in both groups while experiencing fatigue compared to baseline. We did not identify any Day x Group interaction (p>.44) for accuracy data (i.e., iANG and fERR), meaning that the task-specific training did not decrease the impact of fatigue on movement accuracy. However, there was a significant difference between the groups in the evolution of movement speed across days (Day x Group interaction p < .01). The Training

group showed improvement of their speed in both conditions in Day 5 (rested and fatigued), compared to baseline (post-hoc analysis: Training group = Time effect [Condition 1 x Day] p < .01; [Condition 2 x Day] p < .01), while the Control group did not show such improvement between the days (post-hoc analysis Control group = Time effects [Condition 1 x Day/Condition 2 x Day]: p>.20).

## Kinematics

As expected, fatigue impacted kinematics in both groups. During task performance under a fatigued state, both groups showed significant alterations in their initial posture (initial angles), characterized by increased trunk contralateral rotation and extension (Condition effect, p < .01). Participants also used increased shoulder external rotation (Condition effect, p < .01), decreased shoulder elevation combined with a plane of elevation more along the frontal than the sagittal plane (Condition effect, p < .01, Fig 2A) and an increased sternoclavicular

**2.1**

**2.3**

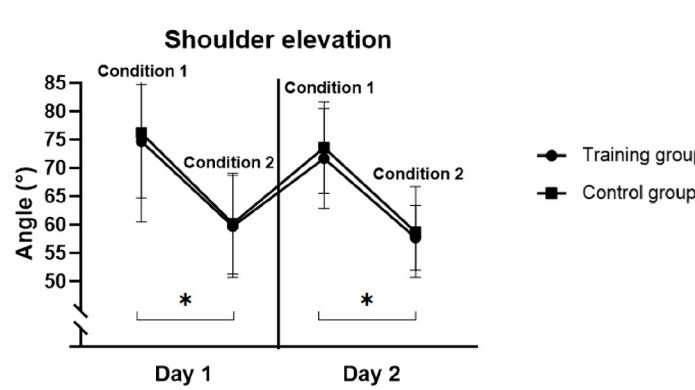

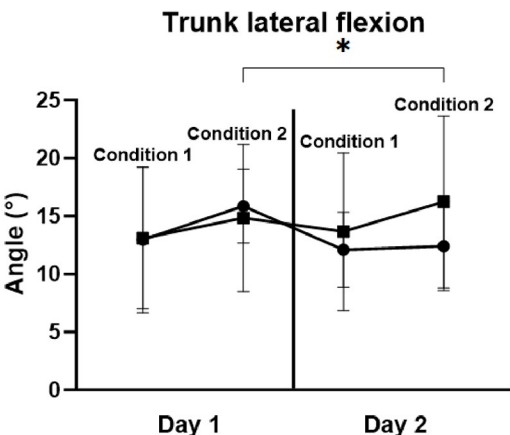

**2.2**

**2.4**

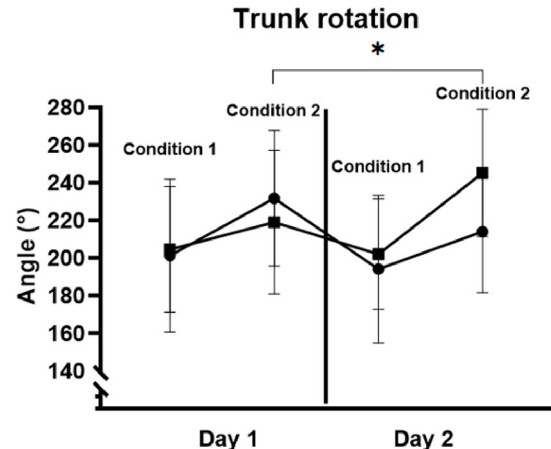

Condition 1=rested; Condition 2=fatigued

**Fig 2. Joints initial angles.** Condition 1 = rested; Condition 2 = fatigued.

elevation (Condition effect, $p < .01$, Fig 2B) compared to the rested condition. Final angles also reflected a change in the reaching strategy under the fatigued condition. Increased trunk contralateral final rotation (Condition effect, $p < .01$) and trunk lateral flexion (Condition effect, $p = .03$) were observed in both groups. Additionally, a higher shoulder external rotation final angle was seen under the fatigued condition, accompanied by an increased use of sterno-clavicular elevation (Condition effect, $p < .01$) and elbow flexion (Condition effect, $p = .03$).

A significant between-group difference was observed in the changes between Day 1 and 5 for trunk compensations (Group x Day interaction $p < .04$ (Fig 2C and 2D). Post-hoc analyses revealed that the Training group showed a reduction in contralateral trunk rotation (Condition 2 x Day: Time effect $p < .01$) and lateral trunk flexion (Condition 2 x Day: Time effect $p < .01$) between the days under fatigue. They were able to maintain a more upright trunk posture when fatigued after their training sessions (Day 5). The Control group's trunk values remained unchanged between the days under the fatigued condition (Condition 2 x Day: time effect $p > .59$). There was also a significant Group x Day interaction ($p = .02$) for trunk contra-lateral rotation final angles, but post-hoc analyses revealed there was no significant Time effect for either the Training or the Control group ($p > .76$).

## Discussion

This study aimed at exploring the effect of task specific training on upper limb adaptations to fatigue during a task involving elevated arm positions [20]. Investigating the potential preventive effects of motor training interventions, such as a task-specific training, represents an initial step toward understanding and offering preventive modalities for musculoskeletal injuries, especially those affecting the shoulder, which are highly prevalent and impose a significant burden worldwide [25]. Some people are frequently exposed to fatiguing tasks, whether in repetitive work or sports, and it is essential to comprehend how to mitigate the risks factors associated with the presence of fatigue.

The initial alterations observed in motor performance and kinematics under fatigue were similar to those noted in a previous study [20]. These adaptations included a decrease in movement accuracy and speed, along with an increased utilization of trunk and sternoclavicular movement. We also noticed reduced shoulder elevation and increased elbow flexion. The task-specific training resulted in improved performance for the Training group, as they exhibited faster reaching movements under fatigue without compromising accuracy. The training also had a significant impact on kinematics, particularly in reducing compensatory trunk movements. The trunk angles, such as initial rotation and lateral flexion angles, decreased following training, even when experiencing the same level of fatigue [EMG signs and perceived level]. These observed trunk adaptations under fatigue were defined as "compensations", assuming that the increase in trunk extension, contralateral flexion, and rotation aimed to achieve the targets with less shoulder elevation [20].

It appears that participants in the Training group relied less on trunk compensation to complete the task after the training period. It can be hypothesized that the kinematic changes observed after the task-specific training reflect an enhanced capacity of the participants to tolerate the physical symptoms of fatigue in main shoulder agonists (e.g., symptoms and signs of fatigue), thereby delaying the onset of trunk compensations. There are potential mechanisms that could explain this capacity, such as the acquisition of a higher motor variability after the training period [26–28]. This may include increased variability in movement patterns, muscle activation, and redistribution of neural drive among agonist muscles [26]. These adaptations have been observed in individuals with extensive experience in repetitive manual tasks [26], and are believed to result from motor learning, and aiming at maintaining performance during

demanding tasks. Other known potential changes in the upper limb subsequent to motor training include enhanced muscle activation in association with improved performance, reduced variability in motor unit discharge, enhanced force steadiness, and improved muscle coordination [27, 28].

Current knowledges suggest there are various mechanisms that may explain the observed changes following task-specific training, primarily related to the development of more efficient motor control. All of these mechanisms might have helped in mitigating the effects of fatigue on the primary shoulder muscles, increasing endurance, and reducing reliance on trunk compensation [26]. While the objective of this exploratory study did not involve investigating underlying mechanisms, such as variability or EMG activity redistribution, it would be interesting for future research to delve into these mechanisms concerning the impact of motor learning and fatigue.

In this study, performance improvement after task-specific training was characterized by enhance speed without compromising accuracy. This improvement could be related to the kinematic changes observed following the training period. Upper limb accuracy (i.e., shoulder movement and hand deviation) depends on an accurate prediction of trunk kinematics and is affected by miscalculated disturbance at the trunk [29]. Assuming that the development of trunk compensations in the fatigued state acted as trunk disturbances, the ability to reduce these compensations after the training period might have helped participants in the training group to improve upper limb performance. It is somewhat surprising that accuracy did not improve after the training period. One possible explanation for that is that the considerable variability (SD in movement trajectory and accuracy) limited the ability to detect any changes in accuracy. This might be related to the high level of difficulty of the task.

## Strengths and limitations

To our knowledge, this is the first study to explore the effect of a task-specific training on motor performance and kinematics while experiencing fatigue. Although it is well-known that fatigue has deleterious effects, such as decreased performance and the potential development of musculoskeletal injuries [4, 7], minimal effort has been made to date to understand how to prevent these effects. Task-specific training is already extensively used in rehabilitation settings for populations with disabilities, such as musculoskeletal injuries and neurological conditions, as they have been shown to be effective for reducing pain [9, 30], improving functional outcomes [9, 30], and even reducing perceived level of fatigue [31]. Exploring the preventive, rather than curative, potential effects of task-specific trainings is a unique contribution to the literature. However, as an exploratory study, we acknowledge the limitations inherent in the study design, and its measurements, which should be mentioned. As we did not identify any study that previously investigated the potential protective effect of a task-specific training on the impact of fatigue, the chosen parameters may not be optimal. It is usually recognized that a high number of repetitions are needed to induce changes in motor control among people with disabilities, with emphasis on conscious attention on the movement performed [9, 31]. Based on these evidence, we included many repetitions performed over a period of three days [32], with conscious focus on movement quality (accuracy) and efficacy (speed). This training exposure seems sufficient, since studies have shown that it is possible to induce motor control changes (i.e., selectively activate different muscle subdivisions to correct scapular posture) after just an hour of training [26]. However, there is not enough literature on this specific subject to confirm that parameters were optimized. It should also be considered that the subjects in this study were young and healthy. Given that factors such as age, gender and the presence of pain can influence motor learning and movement [26, 32], these findings may not apply to

all populations. Looking to the research ahead, it will be interesting to investigate further the mechanisms underlying the changes following a task-specific training and performance in the presence of fatigue, and to determine if these changes can be maintained over time.

## Conclusion

Task-specific training minimized some of the compensations associated with upper-limb reaching in a fatigue-state. Following the 3-day training, performance improved, and participants relied less on trunk compensation to complete the task under fatigue. Task-specific training could help minimize the deleterious effects of fatigue when performing repetitive tasks.

## Supporting information

**S1 Fig. Experimental setup.** Top left: positions of the targets relative to the participant (Target 1 = 90˚ of humeral abduction and 90˚ of external rotation, elbow flexed at 90˚, Target 2 = 90˚ of shoulder abduction, elbow extended, Target 3 = 120˚ of shoulder scaption, elbow extended, Target 4 = 120˚ of shoulder flexion, elbow extended and Target 5 = 140˚ of shoulder flexion, elbow extended. Bottom left: vision of the participant in the virtual reality environment. Right: A left-handed participant in initial position.
(DOCX)

## Acknowledgments

The research team would like to thank Dr Jean Leblond for his help with statistical analyses.

## Author Contributions

**Conceptualization:** Frédérique Dupuis, Jean-Sébastien Roy.

**Data curation:** Jean-Sébastien Roy.

**Formal analysis:** Frédérique Dupuis, Catherine Mercier.

**Funding acquisition:** Jean-Sébastien Roy.

**Investigation:** Frédérique Dupuis, Félix Prud'Homme, Arielle Tougas.

**Methodology:** Frédérique Dupuis, Jean-Sébastien Roy.

**Project administration:** Jean-Sébastien Roy.

**Resources:** Jean-Sébastien Roy.

**Software:** Frédérique Dupuis, Alexandre Campeau-Lecours.

**Supervision:** Jean-Sébastien Roy.

**Validation:** Catherine Mercier.

**Writing – original draft:** Frédérique Dupuis.

**Writing – review & editing:** Félix Prud'Homme, Arielle Tougas, Alexandre Campeau-Lecours, Catherine Mercier, Jean-Sébastien Roy.

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
