## [Decision Letter · Decision Letter 0]

31 Oct 2023

PONE-D-23-24819The effect of a task-specific training on upper limb performance and kinematics while performing a reaching task in a fatigued statePLOS ONE

Dear Dr. Roy,

Thank you for submitting your manuscript to PLOS ONE. After careful consideration, we feel that it has merit but does not fully meet PLOS ONE’s publication criteria as it currently stands. Therefore, we invite you to submit a revised version of the manuscript that addresses the points raised during the review process. Specifically, the reviewers have noted that the rationale for conducting the study and this setup is not always aligned with the introduction and discussion. In addition, more clarity is needed in reporting - potentially met by adding more figures and tables that illustrate the setup and results as well as reviewing the language used. 

We look forward to receiving your revised manuscript.

Kind regards,

Aliah Faisal Shaheen

Academic Editor

PLOS ONE

“This study was funded by the Natural Sciences and Engineering Research Council of Canada (RGPIN-2023-04929). Frederique Dupuis is supported by a scholarship from the Canadian Institute of Health Research (CIHR). JSR is supported by salary awards from Fonds de recherche Québec – Santé (FRQS) and CM holds the Canada Research Chair in Sensorimotor Rehabilitation and Pain and the University Laval Research Chair in Cerebral Palsy.”

Reviewers' comments:

Reviewer's Responses to Questions

**Comments to the Author**

1. Is the manuscript technically sound, and do the data support the conclusions?

Reviewer #1: Partly

Reviewer #2: Partly

2. Has the statistical analysis been performed appropriately and rigorously? 

Reviewer #1: Yes

Reviewer #2: Yes

3. Have the authors made all data underlying the findings in their manuscript fully available?

Reviewer #1: No

Reviewer #2: Yes

4. Is the manuscript presented in an intelligible fashion and written in standard English?

Reviewer #1: No

Reviewer #2: Yes

5. Review Comments to the Author

Reviewer #1: Overall comments. The manuscript is interesting and novel. But it is not clear to me how that reaching task could represent athletic or daily movements. In the introduction, the authors discussed the importance of motor learning to improve the insurance of fatigue and to avoid consequent possible pathologies. But it is not clear to me the practical application of the specific testing, fatigue protocol, and training. It could be better argued in the discussion section, that is in fact too short. It contains a poor description of the obtained results, supported by few comparisons with other studies. Finally, in the manuscript there are some typos and unusual expressions that must be corrected. The authors should pay attention to the manuscript details.

Lines 82-106: I suggest changing the order of the Introduction. Lines 94-106 describe the general effect of fatigue on motor control and performance, and they could be placed before lines 82-91, that are more specific for the upper limbs. Lines 91-93 could be placed before the study aims.

Lines 83-87: The explicit reference to their previous work is unnecessary.

Lines 133-136: I strongly suggest reporting angles only according to the anatomical planes (sagittal, frontal, transversal), so Target 1 and Target 3 position should be expressed as the combination of flexion/extension, adduction/abduction, and rotation of the shoulder and elbow. In addition, I find necessary to provide a figure that shows (or represent) the subject and targets positioning.

Lines 145-147: I suggest providing a reference for Borg RPE Scale interpretation.

Lines 150-153: There is no description of the fatigue protocol. The reference to another paper could not be enough. I strongly suggest adding information about the fatigue protocol, to be exhaustive. In addition, how did the author assess the Borg RPE value? How many times did they asked it to the subjects? Which was the cadence of the evaluations? I found it difficult to understand without the description of the fatigue protocol.

Line 153: The explicit reference to their previous work is confounding, as it seems like they are referring to a previous section of the current manuscript.

Line 183-184: Did the authors measured the duration of training and testing session? How long was, on average, the rest period that subjects considered necessary to avoid fatigue?

Lines 191-192: How was the beginning of the movement computed? Was it defined as the displacement from a baseline? Was it computed by the custom software or obtained from IMUs or from an onset of EMG data?

Lines 194-196: This part is not clear to me. How did the authors measure the angles to assign the scores? I believe more details could help for understanding.

Lines 201-202: There is a lack of information. The reference to the previous study of the authors is not exhaustive.

Lines 204-220: I suggest listing all the measurement methods one after the other. Thus, the authors could place these lines right after Muscles fatigue assessment paragraph.

Lines 187-203: I suggest moving these lines after the measurement methods, before Task specific training.

Line 224: I suggest to explicit the meaning of the NparLD, the software or package the authors used for statistics.

Line 237: All the numbers should be expressed with the same number of digits. In this case: 2.5 ± 1.0.

Lines 256, 258, 263-265, 269-270, 274-275, 277, 285, 287: All the p-values reported appear with different number of digits. The authors should select the number of digits they want to display for p-values and be coherent for the entire manuscript.

Lines 267-268: It would be better to refer to Condition 2 as fatigued condition to improve readability.

Lines 272-273, 279: It is better to introduce the figures in a numeric order (Figure 1.1, Figure 1.2, and so on).

Lines 292: The explicit reference to their previous work is confounding, as it seems like they are referring to a previous section of the current manuscript.

Lines 331-335: The first sentence sounds like a limitation of the study. In addition, which contribute was provided by the measurements of EMG in this protocol? What would have happened if EMG results were significant? Would it become a primary objective?

Lines 338-339: How is the upper limb accuracy affected by miscalculated disturbance at the trunk. An explanation could improve the understanding more than a reference.

Lines 359-360: How did the authors decide the number of tasks, trials, training days, repetitions in the training days, if there is no past literature on the same topic? Did they perform pilot studies?

Reviewer #2: PONE-D-23-24819 Review commends:

This study aimed to investigate the impact of task-specific training on upper extremity kinematics and motor performance under fatigue. Thirty healthy participants were divided into a Training group and Control group, both evaluated for reaching tasks under rested and fatigued conditions. The Training group, after three days of training, exhibited improved speed and reduced trunk compensation when fatigued, indicating that task-specific training can mitigate the effects of fatigue.

1. Additional figures and tables are needed to provide a clearer description of the experimental setup, procedure, and outcome measurements. This would enhance the comprehensibility of the study for the audience.

2. Line 322: Author reference that motor variability may result from motor learning that aimed at maintaining performance during a repetitive demanding task. Have you assessed the difference in the motor variability between Training group and Control group? Did they develop those variability during the 3-day training?

3. Did Training group consistently practice the same reaching movement during the 3-day training? It is noteworthy that there is no Day * Group interaction regarding accuracy, this is surprising, does author have any explanation for this observation?

6. PLOS authors have the option to publish the peer review history of their article (what does this mean?). If published, this will include your full peer review and any attached files.

Reviewer #1: No

Reviewer #2: No

---

## [Author Response · Author response to Decision Letter 0]

14 Nov 2023

Reviewer #1: Overall comments. The manuscript is interesting and novel. But it is not clear to me how that reaching task could represent athletic or daily movements. In the introduction, the authors discussed the importance of motor learning to improve the insurance of fatigue and to avoid consequent possible pathologies. But it is not clear to me the practical application of the specific testing, fatigue protocol, and training. It could be better argued in the discussion section, that is in fact too short. It contains a poor description of the obtained results, supported by few comparisons with other studies. Finally, in the manuscript there are some typos and unusual expressions that must be corrected. The authors should pay attention to the manuscript details.

• Thank you for your comments. They have contributed to improving the quality of the manuscript. In light of the constructive feedback received, we have made several changes in the introduction and the discussion. These changes aim to strengthen the argument for the relevance of this study in relation to the burden of musculoskeletal injury and the risks associated with fatigue. In the clean copy file, we have also made major revisions related to the language and sentence structure to further enhance the manuscript. 

• Please find below the specific changes we made in response to each comment. 

Lines 82-106: I suggest changing the order of the Introduction. Lines 94-106 describe the general effect of fatigue on motor control and performance, and they could be placed before lines 82-91, that are more specific for the upper limbs. Lines 91-93 could be placed before the study aims.

• Thank you for this suggestion. The introduction has been revised, with several changes made to the order and content. We believe this new version of the introduction is now more reader friendly.

Lines 83-87: The explicit reference to their previous work is unnecessary.

• We removed this sentence.

Lines 133-136: I strongly suggest reporting angles only according to the anatomical planes (sagittal, frontal, transversal), so Target 1 and Target 3 position should be expressed as the combination of flexion/extension, adduction/abduction, and rotation of the shoulder and elbow. In addition, I find necessary to provide a figure that shows (or represent) the subject and targets positioning.

• We added a figure of the experimental design (Figure 1, Line 121), as well as a figure of the experimental setup (in supplementary file, Figure S1, Line 136).

Lines 145-147: I suggest providing a reference for Borg RPE Scale interpretation.

• The suggested changes have been made, with references added to lines 145 to 147, and lines 155 to 160. We also added precisions related to the use of the Borg Scale: The participants rated their perceived level of exertion every 30 seconds using the Borg Rating of Perceived Exertion Scale. The three tasks of the fatigue protocol were repeated until the participant reached a perceived level of exertion of at least 8/10, Line 155.

Lines 150-153: There is no description of the fatigue protocol. The reference to another paper could not be enough. I strongly suggest adding information about the fatigue protocol, to be exhaustive. In addition, how did the author assess the Borg RPE value? How many times did they asked it to the subjects? Which was the cadence of the evaluations? I found it difficult to understand without the description of the fatigue protocol.

• We have included additional information to help readers comprehend the fatigue protocol performed and the methods used to evaluate the level of exertion. We added the following sentence, line 151-160: It consisted of three different tasks completed with the dominant arm: 1) manipulating screws on a wooden board for 2 minutes with the shoulders at 45° of flexion; 2) 20 repetitions of arm elevations in the sagittal plane holding a dumbbell; and 3) 20 repetitions of arm elevations in the scapular plane holding the same a dumbbell). The dumbbell used were 0.9 kg (2 pounds) for women and 1.8 kg (4 pounds) for men. Please see the revised version of the ‘’Fatigue protocol’’ section.

Line 153: The explicit reference to their previous work is confounding, as it seems like they are referring to a previous section of the current manuscript.

• We have rephrased this section to prevent any confusion. We used ‘’in a previous study’’ instead of ‘’we previously showed that…’’’. 

Line 183-184: Did the authors measured the duration of training and testing session? How long was, on average, the rest period that subjects considered necessary to avoid fatigue?

• We acknowledge that the duration of both the training and testing sessions was not precisely monitored. Regarding the testing sessions, one trial consisted of 25 reaching movements, which, on average, took 60 seconds to complete. This information has been added to the ‘’Reaching task’’ section of the manuscript, line 135. However, the duration of the training session varied as the requirement for ‘’speed’’ was not consistent throughout. Consequently, it is not feasible to accurately determine the length of the training sessions. Our focus was more on the number of repetitions than on the length of the sessions. Still, we recognize the importance of reporting the total exposition to the task. In future studies, we will ensure to include this information.

• We also did not monitor the exact duration of the rest period for each participant between the trials. To prevent fatigue, we ensured that each participant had a minimum of 2 minutes of rest between trials. However, the actual duration of the rest period varied for each participant based on the time needed to return to a ‘’perceived level of exertion of 0/10’’. This was determined based on their self-reported perceived level of exertion, which was inquired every 30 seconds. We have included this information in the ‘’Task specific training’’ section, line 155-160.

Lines 191-192: How was the beginning of the movement computed? Was it defined as the displacement from a baseline? Was it computed by the custom software or obtained from IMUs or from an onset of EMG data?

• We realized that the confusion arose due to the lack of introduction regarding the standardization of the ‘’initial position’’. To clarify, the initial position consisted of a 5cm radius target that participant were required to reach between movements. This setup allowed us to monitor the position of the hand (with the controller as described) relative to that initial position. We have included this information in the ‘’Reaching task’’ section, line 135-144 on the clean copy.

• We used the data obtained from the controller to track the hand’s position relative to the initial position in the virtual environment, enabling us to determine when the participant initiated the reaching movement.

• The initiation of the movement was computed through a custom-written Matlab program, utilizing the data collected with Unreal Engine. This information is presented in ‘’Measurements and outcomes’’ section. We made few changes to clarify, line 178-181 of the clean copy: Performance was assessed using Unreal Engine, enabling us to track participants’ hand in a three-dimensional space with the controller. Performance data were extracted using custom software written in MATLAB.

Lines 194-196: This part is not clear to me. How did the authors measure the angles to assign the scores? I believe more details could help for understanding.

• We added the following sentence to offer additional details: This angle was calculated using the shortest line between the initial position and the reaching target, and the line corresponding to the initial peak of acceleration. Line 186-188.

Lines 201-202: There is a lack of information. The reference to the previous study of the authors is not exhaustive.

• We have included the following sentence to provide a more comprehensive explanation, avoiding a reference to the previous study: (…) is the summation of the rectangular trapezoids perpendicular to both the ideal trajectory line and the actual trajectory line. Line 194 to 196 of the clean copy. We hope this clarification is helpful. 

Lines 204-220: I suggest listing all the measurement methods one after the other. Thus, the authors could place these lines right after Muscles fatigue assessment paragraph. Lines 187-203: I suggest moving these lines after the measurement methods, before Task specific training. 

• We have made both modifications. Thank you.

Line 224: I suggest to explicit the meaning of the NparLD, the software or package the authors used for statistics.

• We have added the following sentence in the “Statistical analysis” section: (…) a Nonparametric Analysis of Longitudinal Data (NparLD) was conducted using a three-way non-parametric ANOVA for repeated measures (…).

• We have also included information about the software used for statistical analysis.

Line 237: All the numbers should be expressed with the same number of digits. In this case: 2.5 ± 1.0.

• Thank you, we have made the necessary corrections throughout the manuscript.

Lines 256, 258, 263-265, 269-270, 274-275, 277, 285, 287: All the p-values reported appear with different number of digits. The authors should select the number of digits they want to display for p-values and be coherent for the entire manuscript.

• Done. 

Lines 267-268: It would be better to refer to Condition 2 as fatigued condition to improve readability.

• Done. We also made this change throughout the manuscript.

Lines 272-273, 279: It is better to introduce the figures in a numeric order (Figure 1.1, Figure 1.2, and so on).

• Done. 

Lines 292: The explicit reference to their previous work is confounding, as it seems like they are referring to a previous section of the current manuscript.

• We replaced ‘’we previously showed’’ with ‘’a previous study showed’’. 

Lines 331-335: The first sentence sounds like a limitation of the study. In addition, which contribute was provided by the measurements of EMG in this protocol? What would have happened if EMG results were significant? Would it become a primary objective?

• The objective of using EMG in this study was to confirm the presence of fatigue when performing the task in the fatigued state, as experimental sessions were conducted on two separate days. The aim was to ensure that the level of fatigue present in the main agonist muscles remained consistent across both days of investigation. We anticipated minimal changes in EMG fatigue indicators due to the nature of intervention and the criteria for terminating the fatigue protocol. Several factors may have influenced the presence of fatigue between the two days. Consequently, we considered it necessary to monitor the presence of fatigue through EMG sensors. In the event of any disparity between the two days, it would have been challenging to compare them directly, considering the uncertainty regarding the cause of the differences. 

• Given this, if we had observed significant changes in the EMG, it would not have been the primary objective. This is due to the challenge of discerning whether these changes were a result of the training or other influencing factors.

• However, we believe that a few elements contributed to this confusion for the reader, prompting us to make some corrections: 

o In the methods section, muscle fatigue assessment, we changed: ‘’To assess the presence of fatigue, wireless surface (…) for “To monitor the presence of fatigue when performing the reaching task in the fatigued state during the two evaluation sessions (i.e., days), wireless sEMG sensors (Delsys Trigno, USA) were placed on the anterior and middle deltoids and on the upper trapezius of the dominant arm. (…). Line 216 of the clean copy.

o In the muscle assessment results section, we changed this sentence ‘’There was no effect of the Task-specific training program on muscle fatigue (Day x Group interaction p>.43).’’ for the following: ‘’ There was no difference between the days of assessment, or the groups, on muscle fatigue (Time effect and Day x Group interaction p>.43).’’ Line 263-264

o We have revised the section of the discussion identified by the reviewer (line 331-335), as we acknowledge the potential confusion that could arise when discussing these mechanisms in relation to our EMG data. Please refer to the updated section for further details.

Lines 338-339: How is the upper limb accuracy affected by miscalculated disturbance at the trunk. An explanation could improve the understanding more than a reference.

• We have included the following information in the Discussion: Line 348-350 Upper limb accuracy (i.e., shoulder movement and hand deviation) depends on an accurate prediction of trunk kinematics and is affected by miscalculated disturbance at the trunk. Additionally, we have added the following sentence: ‘’These observed trunk adaptations under fatigue were defined as "compensations," assuming that the increase in trunk extension, contralateral flexion, and rotation aimed to achieve the targets with less shoulder elevation.’’ Line 320-322.

Lines 359-360: How did the authors decide the number of tasks, trials, training days, repetitions in the training days, if there is no past literature on the same topic? Did they perform pilot studies?

• We did not conduct a pilot study, but we pre-tested our protocol on members of our team. We selected the parameters based on motor control programs described in previous rehabilitation intervention studies, as they represent the available evidence. However, as you mentioned, these studies are not directly related to our topic but share similar aims. We have recognized this limitation in the “Limitations” section.

Reviewer #2: PONE-D-23-24819 Review commends:

1. Additional figures and tables are needed to provide a clearer description of the experimental setup, procedure, and outcome measurements. This would enhance the comprehensibility of the study for the audience.

• We added a figure of the experimental design (Figure 1, Line 121), as well as a figure of the experimental setup (in supplementary file, Figure S1, Line 136).

2. Line 322: Author reference that motor variability may result from motor learning that aimed at maintaining performance during a repetitive demanding task. Have you assessed the difference in the motor variability between Training group and Control group? Did they develop those variability during the 3-day training?

• No, we have not. To the best of our knowledge, this study was the first to evaluate the potential effects of such a prevention program on fatigue consequences. Hence, we initially did not include any measurement or analysis in the protocol to address this question. Our primary aim was to explore the potential impact on kinematics and motor performance, without delving into detailed mechanistic objectives. Nevertheless, based on these results, it would be interesting to design a study with appropriate measurements to further investigate this hypothesis. We added included elements in the discussion: While the objective of this exploratory study did not involve the investigation of underlying mechanisms, such as variability or EMG activity redistribution, it would be of interest for future research to delve into these mechanisms concerning the impact of motor learning and fatigue. Line 340-342 of the clean copy.

3. Did Training group consistently practice the same reaching movement during the 3-day training? It is noteworthy that there is no Day * Group interaction regarding accuracy, this is surprising, does author have any explanation for this observation?

• Yes, they performed the same reaching movement. One potential explanation could be the high variability in accuracy within each reaching movement for individual participants. The considerable standard deviations might have limited our ability to detect a significant difference. This variability can be related to the task itself, as reaching in a virtual 3D environment can be challenging. We have added the following sentences at the end of this section in the discussion: ‘’ It is somewhat surprising that accuracy did not improve after the training period. One possible explanation for that is the considerable variability (SD in movement trajectory and accuracy) limited the ability to detect any changes in accuracy. This might be related to the high level of difficulty of the task. ‘’ Line 353-356

---

## [Decision Letter · Decision Letter 1]

5 Dec 2023

PONE-D-23-24819R1The effect of a task-specific training on upper limb performance and kinematics while performing a reaching task in a fatigued statePLOS ONE

Dear Dr. Roy,

Thank you for submitting your manuscript to PLOS ONE. After careful consideration, we feel that it has merit but does not fully meet PLOS ONE’s publication criteria as it currently stands. Therefore, we invite you to submit a revised version of the manuscript that addresses the points raised during the review process. Please ensure that you address discrepancies between the versions submitted and make sure that the final version addresses all the comments (previous and new) suggested by the reviewers. 

We look forward to receiving your revised manuscript.

Kind regards,

Aliah Faisal Shaheen

Academic Editor

PLOS ONE

Reviewers' comments:

Reviewer's Responses to Questions

**Comments to the Author**

1. If the authors have adequately addressed your comments raised in a previous round of review and you feel that this manuscript is now acceptable for publication, you may indicate that here to bypass the “Comments to the Author” section, enter your conflict of interest statement in the “Confidential to Editor” section, and submit your "Accept" recommendation.

Reviewer #1: (No Response)

2. Is the manuscript technically sound, and do the data support the conclusions?

Reviewer #1: Yes

3. Has the statistical analysis been performed appropriately and rigorously? 

Reviewer #1: Yes

4. Have the authors made all data underlying the findings in their manuscript fully available?

Reviewer #1: Yes

5. Is the manuscript presented in an intelligible fashion and written in standard English?

Reviewer #1: Yes

6. Review Comments to the Author

Reviewer #1: Overall comments. The introduction has undergone significant enhancement. The study's objective has been explicitly articulated. The discussion is now more focused on the results and has been augmented with pertinent information. I would like to access the supplementary material of the article to view the explanatory images related to the exercises conducted in the study. Additionally, I have observed that the version of the manuscript with Track Changes does not precisely match the one without Track Changes. Some sentences in the text still require changes.

Lines 84-85: I suggest re-formulating this in a more scientific language.

Lines 86-88: I suggest adding a reference here.

Lines 88-90: The sentence is repetitive. You may remove “that reduce the occurrence of such changes in a fatigued state” (lines 89-90).

Lines 98-99: I suggest adding a reference here.

Lines 129-133: I do think it is necessary to express angles using joint anatomical planes, in this case referred to the shoulder or elbow, avoiding expression such as humeral abduction and rotation (Target 1), shoulder scaption (Target 3) by expressing them as combination of sagittal and frontal plane movements.

Line 133: Unfortunately, I cannot see the supplementary file.

Line 136-138: The description of the starting point should be improved. The sentence is not clear and fluent, more confusing than explanatory. The verb “to reach” is repetitive.

Lines 153: Is the closing parenthesis at the end of the sentence an error?

Lines 153-154: I think it is unnecessary to express dumbbell weight in pound, as measurement units must be expressed according to the International System of Units guidelines.

7. PLOS authors have the option to publish the peer review history of their article (what does this mean?). If published, this will include your full peer review and any attached files.

Reviewer #1: No

---

## [Author Response · Author response to Decision Letter 1]

11 Dec 2023

Reviewer #1: Overall comments. The introduction has undergone significant enhancement. The study's objective has been explicitly articulated. The discussion is now more focused on the results and has been augmented with pertinent information. I would like to access the supplementary material of the article to view the explanatory images related to the exercises conducted in the study. Additionally, I have observed that the version of the manuscript with Track Changes does not precisely match the one without Track Changes. Some sentences in the text still require changes.

Lines 84-85: I suggest re-formulating this in a more scientific language.

• Done

Lines 86-88: I suggest adding a reference here.

• Done, the reference added summarizes the current state of knowledge on fatigue.

Lines 88-90: The sentence is repetitive. You may remove “that reduce the occurrence of such changes in a fatigued state” (lines 89-90).

• Removed as suggested.

Lines 98-99: I suggest adding a reference here.

• References 12 to 14 should have been placed after this sentence; consequently, they were relocated accordingly.

Lines 129-133: I do think it is necessary to express angles using joint anatomical planes, in this case referred to the shoulder or elbow, avoiding expression such as humeral abduction and rotation (Target 1), shoulder scaption (Target 3) by expressing them as combination of sagittal and frontal plane movements.

• The planes of movement were added as suggested. However, we believe that some readers would prefer the anatomical angles, so we decided to keep this information within brackets. 

Line 136-138: The description of the starting point should be improved. The sentence is not clear and fluent, more confusing than explanatory. The verb “to reach” is repetitive.

• We made some modifications to improve the description as follow: 

• ‘’ To standardize this starting position, an additional target (5cm radius ball) was positioned in front of the participant, at 90° of humeral elevation (in the sagittal plane, elbow extended). Participants were required to return to this target between each reaching movement to initiate the release of the subsequent target.’’

Lines 153: Is the closing parenthesis at the end of the sentence an error?

• Yes, it is, thank you for noticing.

Lines 153-154: I think it is unnecessary to express dumbbell weight in pound, as measurement units must be expressed according to the International System of Units guidelines.

• We agree, we removed this additional information.

---

## [Decision Letter · Decision Letter 2]

3 Jan 2024

The effect of a task-specific training on upper limb performance and kinematics while performing a reaching task in a fatigued state

PONE-D-23-24819R2

Dear Dr. Roy,

We’re pleased to inform you that your manuscript has been judged scientifically suitable for publication and will be formally accepted for publication once it meets all outstanding technical requirements.

Kind regards,

Aliah Faisal Shaheen

Academic Editor

PLOS ONE

Additional Editor Comments (optional):

Reviewers' comments:

Reviewer's Responses to Questions

**Comments to the Author**

1. If the authors have adequately addressed your comments raised in a previous round of review and you feel that this manuscript is now acceptable for publication, you may indicate that here to bypass the “Comments to the Author” section, enter your conflict of interest statement in the “Confidential to Editor” section, and submit your "Accept" recommendation.

Reviewer #1: All comments have been addressed

2. Is the manuscript technically sound, and do the data support the conclusions?

Reviewer #1: Yes

3. Has the statistical analysis been performed appropriately and rigorously? 

Reviewer #1: Yes

4. Have the authors made all data underlying the findings in their manuscript fully available?

Reviewer #1: Yes

5. Is the manuscript presented in an intelligible fashion and written in standard English?

Reviewer #1: Yes

6. Review Comments to the Author

Reviewer #1: (No Response)

7. PLOS authors have the option to publish the peer review history of their article (what does this mean?). If published, this will include your full peer review and any attached files.

Reviewer #1: No

---

## [Editor Report · Acceptance letter]

12 Jan 2024

PONE-D-23-24819R2 

PLOS ONE

Dear Dr. Roy, 

I'm pleased to inform you that your manuscript has been deemed suitable for publication in PLOS ONE. Congratulations! Your manuscript is now being handed over to our production team.

Kind regards, 

on behalf of

Dr. Aliah Faisal Shaheen 

Academic Editor

PLOS ONE